# Prevalence of phenotypic multi-drug resistant *Klebsiella* species recovered from different human specimens in Ethiopia: A systematic review and meta-analysis

**Biniyam Kijineh[1], Tsegaye Alemeyhu[2]\*, Mulugeta Mengistu[2], Musa Mohammed Ali[2]**

**1** Department of Medical Laboratory Science, Wachemo University College of Medicine and Health Sciences, Hossana, Ethiopia, **2** School of Medical Laboratory Science, Hawassa University College of Medicine and Health Sciences, Hawassa, Ethiopia

\* alemayehutsegaye@ymail.com, tsegayea@hu.edu.et

## Abstract

### Background

Multidrug-resistant (MDR) *Klebsiella* species are among public health important bacteria that cause infections difficult to treat with available antimicrobial agents. Infections with *Klebsiella* lead to high morbidity and mortality in developing countries particularly in patients admitted to the intensive care unit. This systematic review and meta-analysis aimed to determine the pooled prevalence of MDR *Klebsiella* species from different human specimens using studies conducted in Ethiopia from 2018−2022.

### Methods

We have systematically searched online databases such as PubMed/Medline, Google Scholar, Hinari, African journals online, Web of Science, Cochrane, and grey literature (Addis Ababa University and Hawassa University) to identify studies reporting the proportion of MDR *Klebsiella* species in Ethiopia. Published articles were selected based on the Preferred Reporting Item of Systematic Review and Meta-analysis (PRISMA). R-Studio version 4.2.3 was used to conduct pooled prevalence, heterogeneity test, and publication bias. A binary random effect model was used to determine the pooled prevalence. Heterogeneity was checked with the inconsistency index ($I^2$). Publication bias was checked with a funnel plot and Egger test. Sensitivity analysis was conducted with leave-one-out analysis. Joanna Briggs Institute's critical appraisal tool for prevalence studies was used to check the quality of each article.

### Results

In this systematic review and meta-analysis, 40 articles were included in which 12,239 human specimens were examined. Out of the total specimens examined, 721 *Klebsiella* species were isolated and 545 isolates were reported to be MDR *Klebsiella* species. The prevalence of MDR *Klebsiella* species ranged from 7.3%-100% whereas the pooled

**Data Availability Statement:** All relevant data are within the manuscript and its Supporting Information files.

**Funding:** The authors received no specific funding for this work.

**Competing interests:** The authors have declared that no competing interests exist.

prevalence of MDR *Klebsiella* species was 72% (95% CI: 63 − 82%, $I^2$ = 95%). Sub-group analysis based on region revealed the highest prevalence of MDR from Addis Ababa (97%) and the least from the Somali region (33%); whereas sub-group analysis based on the specimen type indicated the highest prevalence was from blood culture specimens 96% and the least was from other specimens (ear and vaginal discharge, and stool) (51%).

## Conclusion

Our finding indicated a high prevalence of MDR *Klebsiella* species found in different human specimens. The prevalence of MDR *Klebsiella* varies across regions in Ethiopia, age, the type of specimens, source and site of infection. Therefore, integrated action should be taken to reduce the prevalence of MDR *Klebsiella* species in regional states and focus on clinical features. Effective infection and prevention control should be applied to reduce the transmission within and outside health care settings.

## Introduction

*Klebsiella* is a member of the Enterobacteriaceae family and is naturally present in the gastrointestinal tract microbiome of healthy humans and animals [1]. This common pathogen causes hospital-acquired surgical wound infections, digestive tract infections, community-onset infections, and nosocomial infection outbreaks [2].

Genus *Klebsiella* includes a variety of species that cause human such as *Klebsiella pneumoniae*, *Klebsiella ozaenae*, *Klebsiella rhinoscleromatis*, *Klebsiella oxytoca* and *Klebsiella aerogenes* [3]. The two main pathotypes of *Klebsiella* species, the multidrug-resistant (MDR) and hyper-virulent (hv) clones, account for a significant fraction of infections [4]. The two branches' strains were regarded as non-overlapping since their respective genetic backgrounds differed [4]. Nevertheless, it has been shown that *Klebsiella* species. can obtain genetic elements and mutations that confer virulence traits and/or antibiotic resistance, which ultimately leads to the emergence of convergent clones known as multidrug-resistant and hyper-virulent (MDR-hv) *Klebsiella* spp [5, 6]. MDR-hv *Klebsiella* species exhibit dual hyper-virulence and antibiotic resistance, and they are believed to be evolving further to generate phenotypically distinct strains [6, 7]. Numerous reports from various continents worldwide have documented a wide range of MDR-hv strains of *Klebsiella* species that have evolved through various mechanisms [8]. MDR-hv *Klebsiella* species have become real superbugs that pose major threats to public health due to the rise in severe infections and the growing lack of effective treatments [9].

In the past decade, antibiotic resistance has become a major global public health concern. Antimicrobial resistance (AMR)-related infections caused 700,000 deaths worldwide each year; by 2050, this number is predicted to increase to 10 million worldwide and 4.2 million in Africa [10]. According to reports, there is a 70% global rate of antibiotic resistance for *Klebsiella*, and the incidence of infection-related mortality ranges from 40% to 70% [4]. In recent years, there has been a growing concern for global public health regarding carbapenem-resistant *Klebsiella* (CRK) and multiple-drug-resistant *Klebsiella* (MDRK). *Klebsiella* species was listed as one of the eight drug-resistant microorganisms in a 2017 WHO report. The advent of multidrug-resistant *Klebsiella* species and limited therapeutic options for infections caused by multidrug-resistant *Klebsiella* species has made treating infections caused by these species more difficult at the moment [11, 12].

Even though there is a clear correlation between colonization and infection, it is unknown what risk factors colonized patients have for infection. It is most likely determined by a combination of bacterial and patient factors whether a patient becomes infected with *Klebsiella*. In a population-level analysis lacking colonization assessment, *Klebsiella* bacteremia was linked to advanced age, male sex, dialysis, chronic liver disease, solid organ transplant, and cancer [13]. Advanced age is linked to the colonization of *Klebsiella* [14]. *Klebsiella* genes and patient characteristics linked to infection with *Klebsiella* as opposed to asymptomatic colonization [15].

Due to their multidrug resistance, *Klebsiella* species is now considered an urgent threat to human health; however, the MDR profile varies amongst countries, even when it comes to the widespread use and misuse of antimicrobial agents [16]. The epidemiology and drug susceptibility pattern of *Klebsiella* species have been studied in various regions of Ethiopia at different times, but the pooled prevalence of MDR *Klebsiella* species in Ethiopia has not been reported. This systematic review and meta-analysis aimed to determine the pooled prevalence of MDR *Klebsiella* species in Ethiopia based on regional states, specimen types, age, source and site of infections.

## Methods

### Literature search strategy

We have systematically searched online databases such as PubMed/Medline, Google Scholar, Hinari, African journals online, Web of Science, Cochrane, and grey literature (Addis Ababa University and Hawassa University) to identify studies reporting the proportion of multi-drug resistant *Klebsiella* species in Ethiopia. Search terms were grouped into four queries (prevalence, antimicrobial susceptibility related term, *Klebsiella spp*, and Ethiopia). The Boolean operator, 'or' (within a query) 'or' (between antibacterial resistance terms), Resistance search terms include "Antibiotic-resistant *Klebsiella*", "Antibiotic susceptible *Klebsiella*", "*Klebsiella* antibiotic sensitivity", "Antimicrobial susceptibility of *Klebsiella*", "Antimicrobial sensitivity of *Klebsiella*", "Antimicrobial resistance of *Klebsiella*", "Antibacterial resistance of *Klebsiella*", "Resistance of *Klebsiella*", "Drug resistance of *Klebsiella*". Studies published from January 1, 2018, to March 3, 2023, were included. We followed the Preferred Reporting Item of Systematic Review and Meta-analysis (PRISMA) to select the articles [17] (S1 Table).

### Data extraction

The data were extracted by three researchers (BK, TA, and MMA) using a standardized and pretested format on April 1–7, 2023. The data abstraction format included the first author, study site, specimen, lab method, sample size, the number of *Klebsiella* species isolated and MDR profile of *Klebsiella* species. Any disagreement on the study to be included/excluded during data extraction between researchers was handled through discussion.

### Quality assessment

Before considering the articles for systematic review and meta-analysis, their quality was checked using nine criteria mentioned in the Joanna Briggs Institute (JBI) critical appraisal tool for prevalence studies [18]. The checklist includes the following questions: Was the sample frame appropriate to address the target population? Were study participants sampled appropriately? Was the sample size adequate? Were the study subjects and the setting described in detail? Was the data analysis conducted with sufficient coverage of the identified sample? Were valid methods used for the identification of the condition? Was the condition measured in a standard, reliable way for all participants? Was there an appropriate statistical analysis?

Was the response rate adequate, and if not, was the low response rate managed appropriately? Articles that scored more than 50% were considered good quality articles and included in the systematic review and meta-analysis [19] (**S2 Table**).

### Eligibility criteria

Articles with a cross-sectional study design, original studies written in English with full-text access, reported MDR *Klebsiella* species, studies conducted only on human specimens, and studies considering both symptomatic and asymptomatic sources of the specimen were included. Studies with quality scores of less than 50% were excluded from the study.

### The outcome of the study

The outcome of this study was the pooled prevalence of MDR *Klebsiella* species recovered from diverse clinical specimens based on the Regional State of Ethiopia, study participants' age, specimen types, source, and site of infection.

### Objective of the study

The systematic review and meta-analysis aimed to determine the pooled prevalence of MDR *Klebsiella* species recovered from diverse human specimens in Ethiopia using studies conducted within the time of 2018–2023.

### Statistical analysis

Data analysis was conducted using R-Studio version 4.2.30 to determine the pooled prevalence, publication bias, and heterogeneity test. A binary random-effects model was used to determine the effect size for the proportion of MDR *Klebsiella*. Subgroup analysis was conducted based on the regional state and specimen types used by the studies. The heterogeneity of the articles was determined using the inconsistency index ($I^2$). Publication bias among the studies was tested with a funnel plot and Eggers, and a *p*-value of < 0.05 was considered statistically significant.

## Results

### Search results

After a thorough search, 98 articles were retrieved, of these, 17 belonged to conference papers, systematic reviews, meta-analyses, letters to the editors, and studies on animal/plan and inanimate objects; they were excluded from the study. After the screening of the title and abstract, 8 articles were duplicates and excluded from the study. Further, 73 articles were screened for full paper, of these 33 articles were excluded from the study because they did not report a quantitative MDR *Klebsiella*, and those articles with Q—score < 50%. Based on this, 10 studies scored 90%, 18 studies scored 80%, 9 studies scored 70%, 2 studies scored 60%, and 1 study scored 50%. Finally, 40 articles were included for both systematic review and meta-analysis (**Fig 1**).

Ethiopia is a country of 11 regional states [Afar, Amhara, Benshangual-Gumuz, Gambela, Harari, Oromia, Somali, Southern Nation's Nationalities, and People's Region (SNNPR), Tigray, Sidama, and Southwest Ethiopia] and two federal cities [Addis Ababa and Dire Dawa]. Article from 6 regions (Amhara, SNNPR, Sidama, Harari, Tigray, Somali) and one federal city Addis Ababa) met the criteria. A total of 40 articles that were published from 2018–2022 identified for the study, including 19 articles from Amhara region [20–38], 7 articles from Addis Ababa [39–45], 5 articles from SNNPR [46–50], 3 articles from Sidama [51–53], 3 articles from Harari [54–56], 2 articles from Tigray [57, 58] and 1 article from Somali [59] (**Fig 2**).

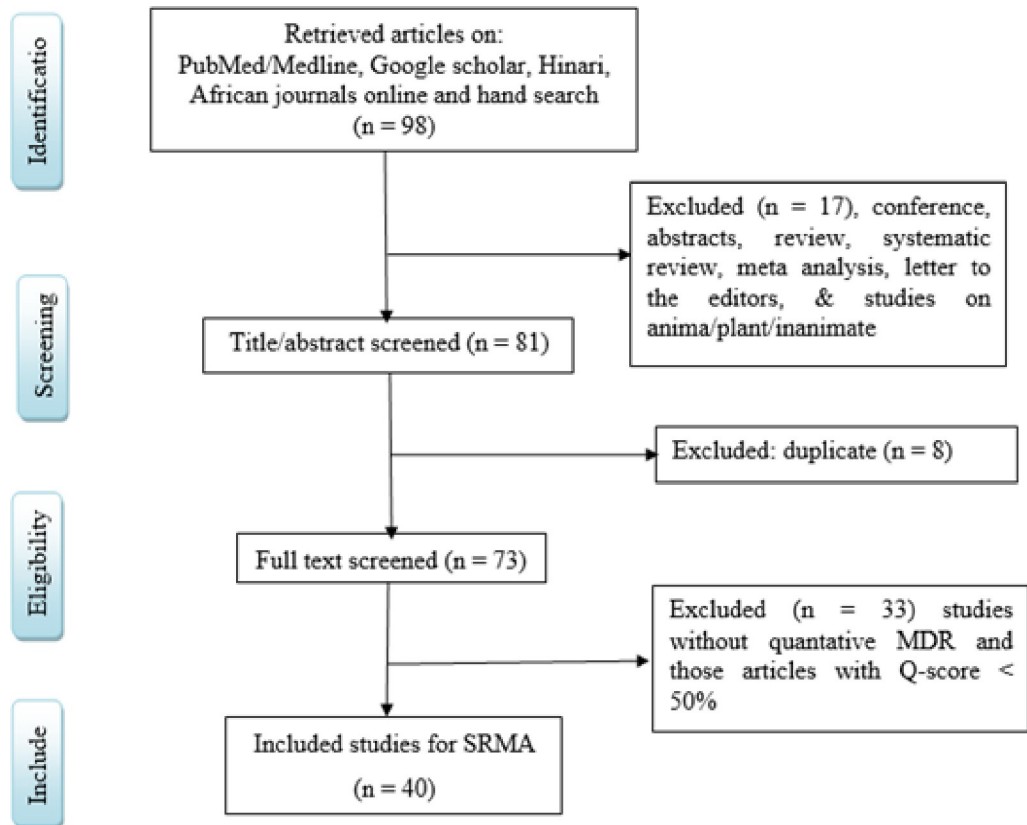

**Fig 1. A PRISMA diagram that shows the flow of article selection study characteristics.**

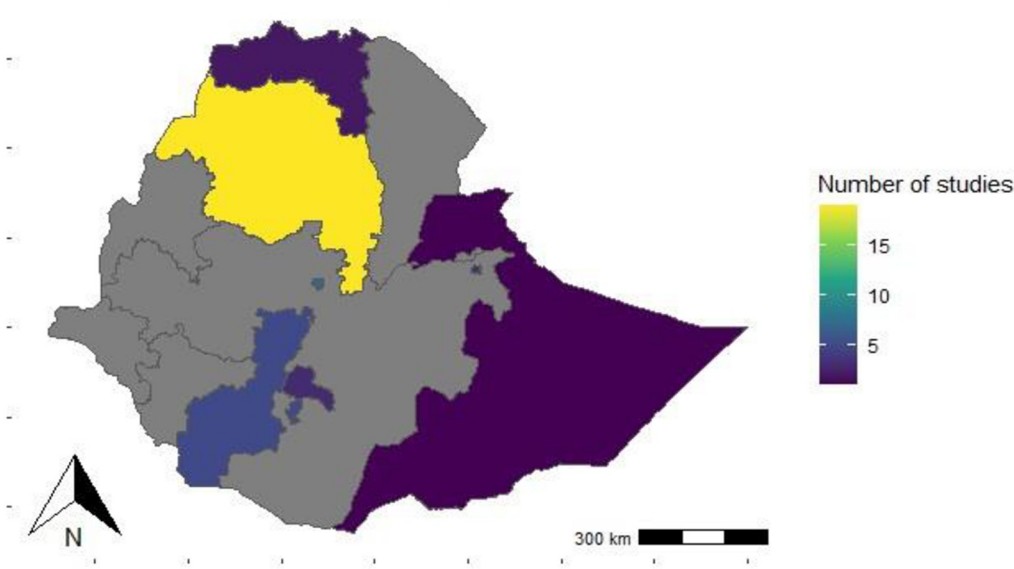

**Fig 2. The map that shows the regional states from which the studies included for the study.**

Concerning articles based on specimen type, most of the articles (21 articles) reported MDR *Klebsiella* from urine [21–23, 25, 27, 30, 33, 38–41, 43, 44, 48–50, 53, 56, 58, 59]. It is followed by seven (7) articles from blood [20, 35, 37, 42, 46, 47, 51], 4 articles from sputum [24, 34, 52, 57], 3 articles from eye swabs [26, 28, 45], 2 articles from body fluid [31, 55], 1 article for each of this specimen stool [32]: vaginal discharge [36] and ear discharge [29]. All the susceptibility testing was conducted with the disk diffusion method. A total of 12,239 patient specimens were analyzed from this, 721 *Klebsiella* were isolated, and of this 545 were reported as MDR (**Table 1**).

## Pooled prevalence of MDR *Klebsiella* species

A total of 12, 239 patients specimen were analyzed, from these 721 *Klebsiella* species were isolated, and 545 isolates were reported as MDR *Klebsiella* species. The review result indicates the prevalence ranged from 7.3%-100% (**Table 1**). According to meta-analysis, the overall pooled prevalence of MDR *Klebsiella* species in Ethiopia was 72% (63–82%) with high heterogeneity ($I^2 = 95\%$, $p < 0.01$) (**Fig 3**).

Because of statistically significant high heterogeneity, we have conducted subgroup analysis based on the regional state, specimen type, source of infection, age of study participants, and site of infection.

## Subgroup analysis

In the subgroup analysis, the prevalence of MDR *Klebsiella* species based on the regions showed that the highest pooled prevalence was from Addis Ababa [97% (95% CI, 93% - 100%), $I^2 = 0\%$, $p = 0.84$] and the least were from Somali [33% (95% CI, 1–91%, $I^2 = $ NA, $p = $ NA. The subgroup analysis based on specimen type the highest was blood culture specimens [96% (92–1.00%), $I^2 = 61\%$, $P = 0.02$] and the least was from other specimens (ear & vaginal discharge, and stool) [51% (95%CI, 0–100%), $I^2 = 96\%$, $P < 0.01$]. Based on the study participants' age the highest and the least proportion of MDR was identified in children [97%(95% CI, 91–100%), $I^2 = 0\%$, $P = 0.53$] and all age groups [61% (95% CI, 44–78%), $I^2 = 91\%$, $P < 0.01$]. The site of infection revealed that BSI had the highest proportion of MDR *Klebsiella* spp. (96% (95% CI, 92–1.00%), $I^2 = 61\%$, $P = 0.02$], while other sites of infection (otitis media, gastrointestinal colonization, and bacterial vaginalis) had the lowest rate (51% (95% CI, 00.0–1.00%), $I^2 = 96\%$, $P < 0.01$). In the other subgroup analysis based on the source of infection, healthcare-associated infections (HAI) were found to have the highest proportion [99% (95% CI, 97–1.00%), $I^2 = 0\%$, $P = 0.053$], which were followed by community-acquired infections (CAI) [68% (95% CI, 56–88%, $I^2 = 87\%$, $P < 0.01$] and both HAI & CAI (67% [95% CI, 44–91%, $I^2 = 98$, $P < 0.01$]) (Table 2).

## Meta-regression

Meta-regression was conducted using regional state, types of specimens, study participants' age, source of infection, site of infection, and publication year to check the source of the heterogeneity but no variables were significantly associated (**Table 3**).

## Publication bias

The funnel plot's asymmetry suggests that there was publication bias, which is corroborated by Egger's test ($P < 0.0001$) (**Fig 4**).

**Table 1. Multi-drug resistant *Klebsiella* species isolated from different human specimens in Ethiopia (2018–2022).**

| SN | Author | Pub year | Study site | Specimen | Study participants Age | Site of Infections | Source of infection | Sample Size | Total-KLB | MDR-KLB | MDR-KLB (%) | Q-score |
|---|---|---|---|---|---|---|---|---|---|---|---|---|
| 1. | Admas *et al* [20] | 2020 | Amhara | Blood | Adult | BSI | Both | 166 | 7 | 7 | 100 | 8 |
| 2. | Adugna [21] | 2021 | Amhara | Urine | All age group | UTI | Both | 422 | 7 | 2 | 28.6 | 9 |
| 3. | Ali *et al* [22] | 2018 | Amhara | Urine | Reproductive age | AUTI | CAI | 358 | 2 | 2 | 100 | 8 |
| 4. | Ameshe [23] | 2020 | Amhara | Urine | All age group | UTI | Both | 385 | 38 | 26 | 68.4 | 8 |
| 5. | Assefa *et al* [24] | 2022 | Amhara | Sputum | Adult | RTI | CAI | 312 | 39 | 37 | 94.9 | 8 |
| 6. | Belete *et al* [25] | 2022 | Amhara | Urine | Children | UTI | CAI | 259 | 5 | 5 | 100 | 8 |
| 7. | Belyhun *et al* [26] | 2018 | Amhara | Eye swab | All age group | OI | CAI | 210 | 8 | 5 | 62.5 | 5 |
| 8. | Girma [27] | 2022 | Amhara | Urine | All age group | UTI | Both | 141 | 8 | 3 | 37.5 | 6 |
| 9. | Haile *et al* [28] | 2022 | Amhara | Eye swab | All age group | OI | CAI | 207 | 5 | 1 | 20 | 8 |
| 10. | Molla *et al* [29] | 2019 | Amhara | Other | All age group | Other* | CAI | 62 | 10 | 2 | 20 | 6 |
| 11. | Oumero *et al* [30] | 2022 | Amhara | Urine | All age group | UTI | CAI | 282 | 12 | 6 | 50 | 9 |
| 12. | Sahle *et al* [31] | 2022 | Amhara | Body fluid | All age group | MI | HAI | 384 | 58 | 57 | 98.3 | 7 |
| 13. | Shenkute *et al* [32] | 2022 | Amhara | Other | All age group | Other* | HAI | 383 | 102 | 101 | 99 | 7 |
| 14. | Tigabu *et al* [33] | 2020 | Amhara | Urine | All age group | AUTI | CAI | 240 | 7 | 3 | 42.9 | 8 |
| 15. | Tilahun *et al.* [34] | 2023 | Amhara | Sputum | All age group | RTI | CAI | 378 | 46 | 43 | 93.5 | 8 |
| 16. | Worku & Tigabu [35] | 2022 | Amhara | Blood | All age group | BSI | CAI | 200 | 8 | 3 | 37.5 | 7 |
| 17. | Yasin *et al* [36] | 2022 | Amhara | Other | Adult | Other* | CAI | 214 | 7 | 2 | 28.6 | 8 |
| 18. | Molla *et al* [37] | 2021 | Amhara | Blood | Neonate | BSI | CAI | 412 | 57 | 53 | 93 | 7 |
| 19. | Fenta *et al* [38] | 2020 | Amhara | Urine | Children | UTI | CAI | 299 | 7 | 6 | 85.7 | 7 |
| 20. | Bizuayehu *et al* [39] | 2022 | Addis Ababa | Urine | Adult | UTI | HAI | 220 | 9 | 9 | 100 | 7 |
| 21. | Bizuwork *et al* [40] | 2021 | Addis Ababa | Urine | Reproductive age | AUTI | CAI | 283 | 8 | 8 | 100 | 8 |
| 22. | Duffa *et al* [41] | 2018 | Addis Ababa | Urine | Children | UTI | Both | 384 | 17 | 15 | 88.2 | 8 |
| 23. | Sherif [42] | 2022 | Addis Ababa | Blood | Neonate | BSI | Both | 400 | 37 | 36 | 97.3 | 8 |
| 24. | Wabe *et al* [43] | 2020 | Addis Ababa | Urine | Reproductive age | UTI | CAI | 290 | 4 | 3 | 75 | 9 |
| 25. | Yenehun *et al* [44] | 2022 | Addis Ababa | Urine | Adult | UTI | CAI | 225 | 3 | 3 | 100 | 8 |
| 26. | Woreta *et al* [45] | 2022 | Addis Ababa | Eye swab | All age group | OI | CAI | 323 | 6 | 6 | 100 | 8 |
| 27. | Ameya *et al* [46] | 2022 | SNNPR | Blood | Children | BSI | Both | 238 | 4 | 4 | 100 | 9 |
| 28. | Birru [47] | 2021 | SNNPR | Blood | Adult | BSI | Both | 225 | 4 | 3 | 75 | 9 |
| 29. | Hantalo *et al* [48] | 2022 | SNNPR | Urine | Adult | UTI | CAI | 217 | 3 | 3 | 100 | 9 |
| 30. | Mitiku *et al* [49] | 2022 | SNNPR | Urine | Adult | UTI | CAI | 422 | 39 | 22 | 56.4 | 9 |
| 31. | Oumer *et al* [50] | 2022 | SNNPR | Urine | Adult | UTI | HAI | 231 | 9 | 9 | 100 | 7 |
| 32. | Alemayehu *et al* [51] | 2019 | Sidama | Blood | Children | BSI | HAI | 939 | 21 | 21 | 100 | 8 |
| 33. | Gebre [52] | 2021 | Sidama | Sputum | Adult | RTI | CAI | 406 | 36 | 12 | 33.3 | 9 |
| 34. | Mechal *et al* [53] | 2022 | Sidama | Urine | Adult | UTI | CAI | 395 | 14 | 9 | 64.3 | 9 |
| 35. | Ejerssa *et al* [54] | 2021 | Harari | Urine | Reproductive age | UTI | CAI | 200 | 5 | 3 | 60 | 8 |
| 36. | Tolera *et al* [55] | 2018 | Harar | Body fluid | All age group | MI | HAI | 394 | 4 | 2 | 50 | 8 |
| 37. | Marami D [56] | 2019 | Harar | Urine | Adult | UTI | CAI | 350 | 15 | 4 | 26.7 | 9 |

(*Continued*)

**Table 1.** (Continued)

| SN | Author | Pub year | Study site | Specimen | Study participants Age | Site of Infections | Source of infection | Sample Size | Total-KLB | MDR-KLB | MDR-KLB (%) | Q-score |
|---|---|---|---|---|---|---|---|---|---|---|---|---|
| 38. | Adhanom *et al* [57] | 2019 | Tigray | Sputum | Adult | RTI | Both | 252 | 41 | 3 | 7.3 | 7 |
| 39. | Gebremariam *et al* [58] | 2019 | Tigray | Urine | Adult | UTI | CAI | 341 | 6 | 5 | 83.3 | 8 |
| 40. | Negussie *et al* [59] | 2018 | Somali | Urine | Reproductive age | UTI | CAI | 190 | 3 | 1 | 33.3 | 7 |
| Total | | | | | | | | 12,239 | 721 | 545 | | 314 |

**Keynote**: Other (vaginal, ear discharge, & stool), Other* (Otitis media, gastrointestinal colonization, and bacterial vaginalis), UTI- urinary tract infection, BSI—bloodstream infection, OM-otitis medium, OI- ocular infection, AUTI- asymptomatic urinary tract infection, RTI-respiratory tract infection, MI-multiple infection, HAI- hospital-acquired infection, and CAI- community-acquired infection.

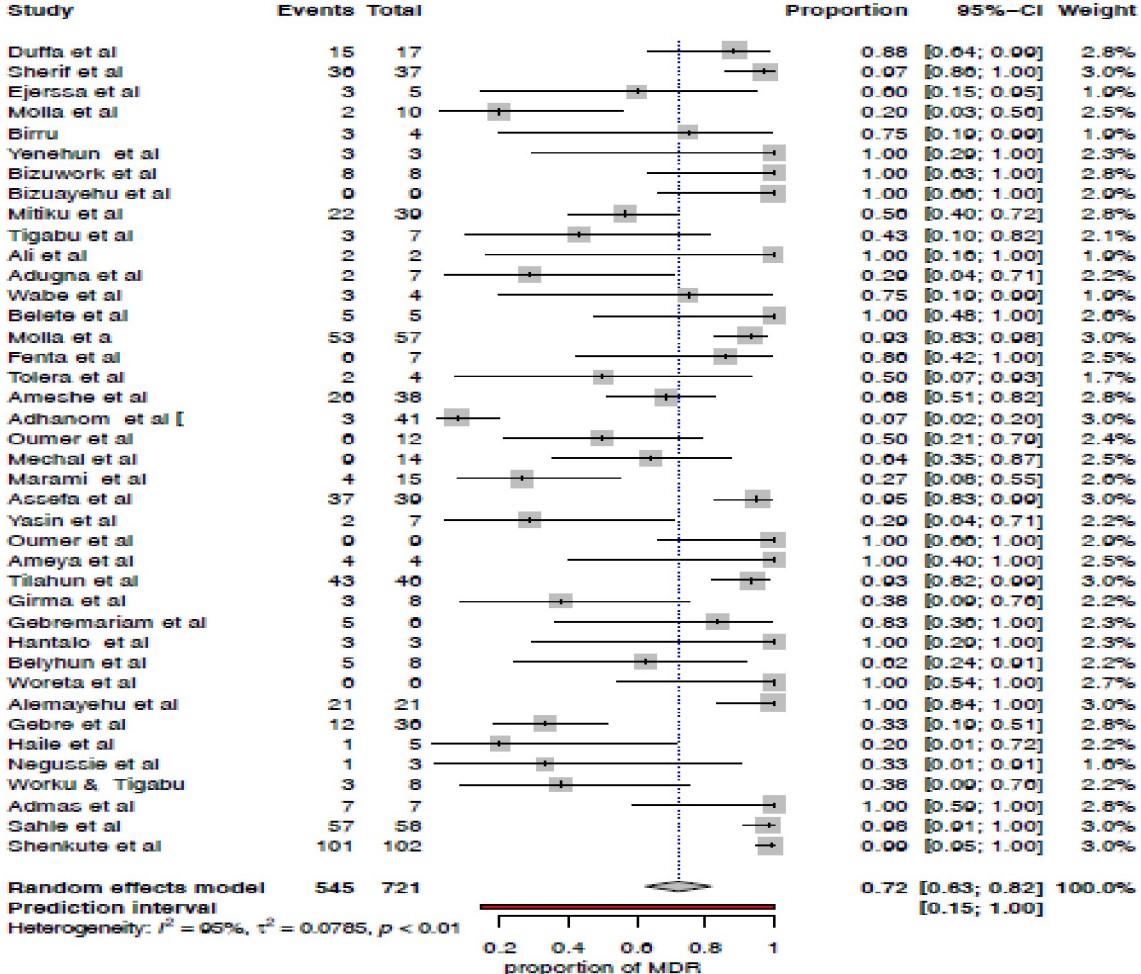

**Fig 3. The forest plot that indicates the pooled prevalence of MDR *Klebsiella* species.**

**Table 2. Subgroup analysis based on regional state, specimen types, study participants' age, site of infection and source of infection.**

| Subgroup | No of studies | MDR [95%CI] | Heterogeneity | |
|---|---|---|---|---|
| | | | *P*-value | $I^2$ (%) |
| **Regional state** | | | | |
| Addis Ababa | 7 | 0.97 [0.93,1.00] | = 0.84 | 0 |
| Harari | 3 | 0.39[0.16, 0.62] | = 0.34 | 8 |
| Amhara | 19 | 0.69[0.55,0.83] | < 0.01 | 89 |
| SNNPR | 5 | 0.69[0.66,1.00] | < 0.01 | 80 |
| Tigray | 2 | 0.44[0.00,1.00] | < 0.01 | 96 |
| Sidama | 3 | 0.66[0.27,1.00] | < 0.01 | 97 |
| **Specimen type** | | | | |
| Urine | 21 | 0.73[0.63,0.85] | < 0.01 | 79 |
| Blood | 7 | 0.96[0.92,1.00] | = 0.02 | 61 |
| Body fluid | 2 | 0.81[0.35,1.00] | = 0.05 | 73 |
| Sputum | 4 | 0.57[0.14,1.00] | < 0.01 | 97 |
| Eye swab | 3 | 0.63[0.17,1.00] | < 0.01 | 88 |
| Other | 3 | 0.51[0.00–100] | <0.01 | 96 |
| **Study participants age** | | | | |
| All age groups | 14 | 0.61[0.44–0.78] | < 0.01 | 91 |
| Adult | 14 | 0.69[0.51–0.87] | < 0.01 | 97 |
| Neonates | 2 | 0.96[0.92–1.00] | = 0.32 | 0 |
| Children | 5 | 0.97[0.91–1.00] | = 0.58 | 0 |
| Reproductive age women | 5 | 0.80[0.57–1.00] | = 0.07 | 53 |
| **Site of Infection** | | | | |
| UTI | 18 | 0.72[0.60–0.84] | < 0.01 | 79 |
| BSI | 7 | 0.96[0.92–1.00] | = 02 | 61 |
| RTI | 4 | 0.57[0.14–1.00] | < 0.01 | 99 |
| OI | 3 | 0.63[0.17–1.00] | < 0.01 | 88 |
| ASUTI | 3 | 0.82[0.46–1.00] | = 0.02 | 75 |
| MI | 2 | 0.81[0.35–1.00] | = 0.03 | 73 |
| Other* | 3 | 0.51[0.00–1.00] | < 0.01 | 96 |
| **Source of infection** | | | | |
| HAI | 6 | 0.99[0.97–1.00] | = 0.53 | 0 |
| CAI | 25 | 0.68[0.56–0.88] | < 0.01 | 87 |
| Both HAI & CAI | 9 | 0.67[0.44–0.91] | < 0.01 | 98 |

**Table 3. Shows the meta-regression based on regional states, types of specimens, study participants' age, source of infection, site of infection and publication year.**

| Variables | DF | Coefficient | p-value |
|---|---|---|---|
| Regional state | 7 | 11.88 | 0.11 |
| Types of specimens | 7 | 10.56 | 0.16 |
| Study participants age | 8 | 7.03 | 0.53 |
| Source of infection | 2 | 5.19 | 0.08 |
| Site of infection | 8 | 10.42 | 0.24 |
| Publication year | 1 | 2.80 | 0.10 |

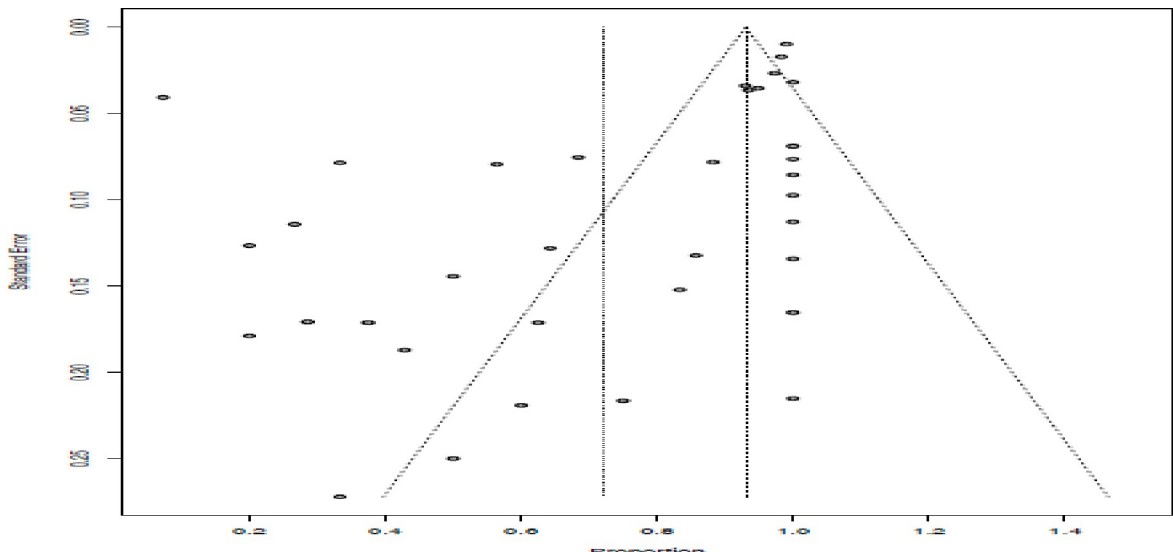

**Fig 4. A funnel plot that indicates the publication bias of the studies.**

### Sensitivity analysis

During sensitivity analysis using leave-one-out analysis revealed that the pooled prevalence range of MDR *Klebsiella* remains unchanged (**Table 4**).

## Discussion

In hospitalized patients, as well as those in nursing homes and other healthcare facilities, *Klebsiella* species can cause a range of infections, including bloodstream infections, pneumonia, and urinary tract infections more seriously in immunocompromised patients [60]. *Klebsiella* are often resistant to multiple antibiotics. Evidence implicates plasmids as the primary source of the resistance genes. *Klebsiella* species with the ability to produce extended-spectrum beta-lactamases (ESBL) are resistant to virtually all beta-lactam antibiotics, except carbapenems [1]. Developing countries are particularly severely affected by these resistant bacteria if appropriate action is not taken right away. Our study attempted to compile the dispersed data in Ethiopia using a systematic review and meta-analysis because it is one of the nations that has experienced improper antibiotic consumption.

In our finding, the pooled prevalence of MDR *Klebsiella species* was estimated to be 72% (63–82%). Our finding is higher than the study conducted at a global level estimates of MDR *K. pneumoniae* 32.8% [61] and Nepal 55% [62]. This difference may be due to their study was conducted on only MDR *K. pneumoniae* from hospital-acquired infection isolates and our study included all *Klebsiella* species on both CAI and HAI. Our finding is comparable with systematic review and meta-analysis from Nepal 64% [63] which was conducted on MDR *K. pneumoniae* from clinical isolates even if our study is on overall *Klebsiella* species. It is also comparable with a study from Ethiopia 68% [64] which was conducted with one health approach from human, animal, and environmental samples. These similarities indicate MDR *Klebsiella* species is circulating anywhere and fighting AMR with one health approach is much more important nowadays.

Globally, AMR has been documented and has the potential to spread quickly. Along with the usage of antibiotics, there are differences in the distribution of resistance and the number of infections around the world. There is a significant knowledge gap regarding the magnitude

**Table 4. Shows sensitivity analysis with the leave-one-out analysis.**

| S. No | Study Omitted | MDR [95%CI] | tau^2 |
|---|---|---|---|
| 1. | Admas *et al* [20] | 0.75[0.72; 0.78] | 4.11 |
| 2. | Adugna [21] | 0.74[0.71; 0.78] | 3.86 |
| 3. | Ali *et al* [22] | 0.76[0.72; 0.79] | 4.13 |
| 4. | Ameshe [23] | 0.76 [0.73; 0.79] | 3.85 |
| 5. | Assefa *et al* [24] | 0.76[0.72; 0.79] | 4.12 |
| 6. | Belete *et al* [25] | 0.76[0.72; 0.79] | 3.93 |
| 7. | Belyhun *et al* [26] | 0.75[0.72; 0.78] | 3.83 |
| 8. | Girma [27] | 0.75[0.72; 0.78] | 3.82 |
| 9. | Haile *et al* [28] | 0.77[0.73; 0.80] | 4.14 |
| 10. | Molla *et al* [29] | 0.76[0.73; 0.79] | 4.07 |
| 11. | Oumero *et al* [30] | 0.76 [0.72; 0.79] | 3.96 |
| 12. | Sahle *et al* [31] | 0.76[0.73; 0.79] | 3.96 |
| 13. | Shenkute *et al* [32] | 0.76[0.72; 0.79] | 4.12 |
| 14. | Tigabu *et al* [33] | 0.75[0.72; 0.78] | 3.89 |
| 15. | Tilahun *et al.* [34] | 0.74[0.71; 0.77] | 4.06 |
| 16. | Worku & Tigabu [35] | 0.76[0.72; 0.79] | 4.10 |
| 17. | Yasin *et al* [36] | 0.76[0.73; 0.79] | 4.10 |
| 18. | Molla *et al* [37] | 0.76[0.73; 0.79] | 4.17 |
| 19. | Fenta *et al* [38] | 0.797[0.77; 0.82] | 3.44 |
| 20. | Bizuayehu *et al* [39] | 0.76[0.73; 0.79] | 4.11 |
| 21. | Bizuwork *et al* [40] | 0.76[0.73; 0.79] | 4.16 |
| 22. | Duffa *et al* [41] | 0.77[0.73; 0.80] | 3.92 |
| 23. | Sherif [42] | 0.75[0.71; 0.78] | 3.99 |
| 24. | Wabe *et al* [43] | 0.76[0.73; 0.79] | 3.96 |
| 25. | Yenehun *et al* [44] | 0.75[0.72; 0.78] | 3.82 |
| 26. | Woreta *et al* [45] | 0.76[0.72; 0.79] | 3.91 |
| 27. | Ameya *et al* [46] | 0.74[0.71; 0.78] | 4.04 |
| 28. | Birru [47] | 0.76[0.73; 0.79] | 4.03 |
| 29. | Hantalo *et al* [48] | 0.76[0.72; 0.79] | 4.11 |
| 30. | Mitiku *et al* [49] | 0.76[0.72; 0.79] | 3.93 |
| 31. | Oumer *et al* [50] | 0.76[0.73; 0.79] | 4.15 |
| 32. | Alemayehu *et al* [51] | 0.75 [0.72; 0.78] | 3.87 |
| 33. | Gebre [52] | 0.75[0.72; 0.78] | 3.68 |
| 34. | Mechal *et al* [53] | 0.78[0.75; 0.81] | 3.99 |
| 35. | Ejerssa *et al* [54] | 0.76 [0.73; 0.79] | 3.89 |
| 36. | Tolera *et al* [55] | 0.76[0.73; 0.79] | 4.02 |
| 37. | Marami D [56] | 0.76[0.73; 0.79] | 4.03 |
| 38. | Adhanom *et al* [57] | 0.75[0.72; 0.78] | 3.85 |
| 39. | Gebremariam *et al* [58] | 0.74[0.70; 0.77] | 3.75 |
| 40. | Negussie *et al* [59] | 0.72[0.68; 0.75] | 3.59 |

of AMR in the world. Particularly the knowledge gap is common in low- and middle-income countries that lack systems to gather data on infections and antibiotic-resistant bacteria [65]. In our systematic review and meta-analysis, a subgroup analysis was conducted based on the regional states, type of specimens, study participants' age, source of infection (CAI vs HAI), and types of infection as the data is heterogenous ($I^2$ = 95%).

The MDR *Klebsiella* species prevalence varied among the regional states, according to the subgroup analysis. Addis Ababa had the highest pooled prevalence, at 97% (93–100%), while Somali had the lowest, at 33% (1–91%). The number of studies included the nature of study participants, and the sample size could all be contributing factors to this variation. One possible explanation could be variations in access to clean water, hygienic practices, and sanitation for both humans and animals; infection prevention and control measures at homes, hospitals, and farms; availability of reasonably priced vaccines; access to diagnostics and medications; and awareness and knowledge regarding antibiotic resistance and appropriate use of antibiotics, which are the primary risk factors for the development of antibiotic resistance as outlined by the World Health Organisation [66].

The sub-group analysis based on the types of specimens shows that the blood culture had the highest prevalence of MDR Klebsiella spp., at 96% (92–100%), while the other specimen cultures (stool, ear discharge, and vaginal discharge) had the lowest prevalence, at 51% (0.0–100). This can be explained by the fact that the latter specimen came from infections acquired in the community, while the former came from infections acquired in hospitals. Unless multiple blood culture bottles are used to rule out infection versus contamination [67], the other possibility may be related to the risk of contamination of blood culture from the environment by *Klebsiella* species from hospital-acquired infections. Some microbiology laboratories in developing countries use single bottles for pediatric blood cultures due to supply shortages [68].

Every individual, at any stage of life, is susceptible to AMR. Individuals who are undergoing medical treatment or have compromised immune systems are frequently more vulnerable to infection [69]. According to global estimates for 2019, children bear a disproportionate share of the burden of death: of the 1.27 million deaths directly attributable to AMR, 254,000 occurred among those under the age of five, accounting for approximately 20% of all deaths. This is the equivalent of one child passing away almost every two minutes. More than 99 percent of those 254,000 kids are from low- and middle-income countries, and more than half of them pass away in their first month of life. Stated differently, 900 children in high-income countries (HICs) and nearly 253,000 in low- and middle-income countries (LMICs) died directly from AMR [70]. Our study also encompassed a subgroup analysis based on the age of the study subjects, which revealed that the lower age group (children and neonates) had a higher prevalence of MDR *Klebsiella* species (97% (91–100%) vs 96% (92–100%)). Several facets of this growing problem are unique to children. Without immediate action, we are at risk of entering a post-antibiotic era in which common infections could once again be fatal in such a rapidly changing environment. This is especially true for children and babies less than a week old, as their gastrointestinal tracts contain populations of MDR bacteria, most likely because of exposure to mother and environmental bacteria during and right after delivery [71].

In our review, a higher MDR *Klebsiella* species were identified from HAI 97% (97–100%). This might be because staying in a hospital or other healthcare facility increases the risk of contracting an antibiotic-resistant infection. Antibiotic exposure is frequent for patients in these facilities, and they are often prone to a lot of hands-on care which may expose them to resistant bacteria from either medical equipment or health professionals. Furthermore, hospitals are more likely than communities to harbour the majority of resistant bacteria [72].

In general, our study has the following limitations. One of the study's limitations is that it only looked at disc diffusion; it did not find a study which used minimum inhibitory concentration (MIC) or molecular techniques. It only used patient specimens, focused on studies conducted between 2018 and 2022, and failed to determine whether certain specimens, such as urine, faeces, sputum, and discharges, were from infection or colonization. Our study considered the report of MDR *Klebsiella* spp. as one of the eligibility criteria but not the definition of MDR because the studies used different definitions and some included simply the MDR value.

## Conclusion

Our study revealed that the prevalence of MDR *Klebsiella* species was high in Ethiopia. The subgroup analyses elaborated on the proportion of MDR *Klebsiella* spp. that was different between regional states, types of specimens, study participants' ages, sources and types of infection. Therefore, integrated action should be taken to reduce the rates of multi-drug-resistant *Klebsiella* to regional states, age, and focus on clinical features of patients. Standard precautions should be applied to reduce the transmission of MDR *Klebsiella* spp in hospital and out of the hospital.

## Supporting information

**S1 Checklist. Human participants research checklist.**
(DOCX)

**S1 Table. PRISMA 2020 checklist.**
(DOCX)

**S2 Table. The quality score of each study based on Joanna Briggs Institute (JBI).**
(DOCX)

## Author Contributions

**Conceptualization:** Biniyam Kijineh, Tsegaye Alemeyhu, Musa Mohammed Ali.

**Data curation:** Biniyam Kijineh, Tsegaye Alemeyhu, Musa Mohammed Ali.

**Formal analysis:** Biniyam Kijineh, Tsegaye Alemeyhu, Musa Mohammed Ali.

**Funding acquisition:** Biniyam Kijineh, Tsegaye Alemeyhu, Musa Mohammed Ali.

**Investigation:** Biniyam Kijineh, Tsegaye Alemeyhu, Musa Mohammed Ali.

**Methodology:** Biniyam Kijineh, Tsegaye Alemeyhu, Musa Mohammed Ali.

**Project administration:** Biniyam Kijineh, Tsegaye Alemeyhu, Musa Mohammed Ali.

**Resources:** Biniyam Kijineh, Tsegaye Alemeyhu, Musa Mohammed Ali.

**Software:** Biniyam Kijineh, Tsegaye Alemeyhu, Musa Mohammed Ali.

**Supervision:** Biniyam Kijineh, Tsegaye Alemeyhu, Musa Mohammed Ali.

**Validation:** Biniyam Kijineh, Tsegaye Alemeyhu, Musa Mohammed Ali.

**Visualization:** Biniyam Kijineh, Tsegaye Alemeyhu, Musa Mohammed Ali.

**Writing – original draft:** Biniyam Kijineh, Tsegaye Alemeyhu, Mulugeta Mengistu, Musa Mohammed Ali.

**Writing – review & editing:** Biniyam Kijineh, Tsegaye Alemeyhu, Mulugeta Mengistu, Musa Mohammed Ali.

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
