## [Decision Letter · Decision Letter 0]

26 Oct 2023

PONE-D-23-29785Prevalence of Multi-drug resistant Klebsiella recovered from clinical specimens in Ethiopia: A Systematic Review and Meta-analysisPLOS ONE

Dear Dr. Begashaw,

Thank you for submitting your manuscript to PLOS ONE. After careful consideration, we feel that it has merit but does not fully meet PLOS ONE’s publication criteria as it currently stands. Therefore, we invite you to submit a revised version of the manuscript that addresses the points raised during the review process.

We look forward to receiving your revised manuscript.

Kind regards,

Melaku Ashagrie Belete, MSc.

Academic Editor

PLOS ONE

 [The funders had no role in study design, data collection and analysis, decision to publish, or preparation of the manuscript.]. 

Additional Editor Comments:

The authors sought to assess the prevalence of MDR Klebsiella recovered from clinical specimens in Ethiopia through a Systematic Review and Meta-analysis approach using data from published articles. Such Meta-analysis studies provide a comprehensive data and are of paramount significance for policy makers. In addition to the revision according to the comments by the reviewers, I would like you to address the following concerns:

Generally, while the objective of this study is interesting, the reporting is insufficient (for example, there are major concerns with respect to the study outcome definition) and there are major concerns regarding the data analysis and discussion write-up. The conclusion drawn from the study is simply descriptive and does not provide recommendations for policy makers on how to cope up with the alarmingly high MDR pooled prevalence. My point-by-point comments that must be addressed in the revision are articulated below:

- The authors are totally unclear regarding the outcome. The abstract and introduction leave the reader in the dark with regard to the exact study outcome. In the methods section the outcome is then reported as MDR Klebsiella species, but without further detail. If the authors focused on infection, they should clearly explain why they included studies on carriage also (for example, see Ali IE et al. 2018, Tigabu A et al. 2020, Bizuwork K et al. 2021, Wabe et al. 2020 and Gebremariam G et al. 2019 on asymptomatic UTI carriage). Additionally, the authors do not explain the criteria for infection or provide any outcome definitions – individual study outcomes are also not included in the table.

- There are very strong concerns regarding the pooling of different specimen types (blood, urine, ear discharge, eye swabs, body fluids etc..) and populations (neonates, adults, HIV patients, pregnant women, cancer patients, etc.) and, of course this leads to huge discrepancies in prevalence.

- There are several grammatical errors throughout the manuscript that need corrections.

- Have you considered the definitions each authors used to declare an isolate as MDR? You didn’t put the definition of MDR in the inclusion criteria either. I have seen some differences in MDR definitions among included studies and this may raise doubt in the credibility of the findings of this meta-analysis.

- The inclusion of a single group of study (only 1 study from Somalia) is inappropriate. The authors should perform sub-group analysis for groups composed of 2 or more studies.

- Search strategies: why did you choose these specific databases? Some of the reported databases have very high overlaps.

- Please include all search strategies in detail in the supplement in line with PRISMA recommendations (database, date searched, key words, number of articles retrieved etc).

- Please include the university repositories used and the Google search line as a supplement.

- Eligibility criteria: human specimens of what material? It seems you included any material and made no distinction between infection and carriage.

- Quality assessment: Did you exclude all other studies with a score <50%? (It seems like you did according to Figure 1). If so, please report them in the supplement. There were no studies with high risk of bias? It would be helpful if the authors can add a legend for the domains of the JBI scoring system.

- The statistical data analysis used in this study are not enough. The authors claimed that there was high level of heterogeneity however they did not try to assess the cause of the heterogeneity deploying different analysis methods such as sensitivity analysis and meta-regression. Similarly, Egger’s test statistics and Trim and Fill analysis were not carried out to provide indications of the reliability of the estimate in relation to publication bias. In fact, the authors only used funnel plot to check publication bias. However, this method is not quantitative and leads to inaccurate interpretations when used alone. The absence of these significant statistical analysis tools simply makes this Meta-analysis a descriptive report.

- No need to write ‘Disk diffusion’ in Table 1 as all studies used disk diffusion. The authors should mention it in the Study characteristics section.

- Line 153-157 “However, the meta-analysis result shows the overall pooled prevalence of MDR Klebsiella species in Ethiopia is 72% (63-82%) with a low heterogeneity (I2 = 0.05%) with a significance value of p < 0.01(Figure 2). Since the data was highly heterogeneous (I2 = 95%) subgroup analysis was conducted based on the regional state and specimen type”. These statements are very contradicting and need explanation.

- I don’t think enough subgroup analysis were carried out in order to assess the source of heterogeneity.

- The Discussion section is poorly written. In fact, it is incorrectly written. The authors must discuss their findings with similar Meta-analysis conducted elsewhere. However, the whole discussion is written by comparing findings of the Meta-analysis with individual studies from different countries and this is totally wrong. Findings of Meta-analysis should be compared with other Meta-analysis reports. This mistake jeopardizes the scientific significance of this paper.

- Was there any limitation in this Meta-analysis? If so, it should be well written at the end of the Discussion section.

- The conclusions lack recommendations so that the outputs of this meta-analysis will be helpful in mitigating prevention and control strategies against MDR Klebsiella spp.

Reviewers' comments:

Reviewer's Responses to Questions

**Comments to the Author**

1. Is the manuscript technically sound, and do the data support the conclusions?

Reviewer #1: Yes

Reviewer #2: Yes

Reviewer #3: Partly

2. Has the statistical analysis been performed appropriately and rigorously? 

Reviewer #1: Yes

Reviewer #2: Yes

Reviewer #3: Yes

3. Have the authors made all data underlying the findings in their manuscript fully available?

Reviewer #1: Yes

Reviewer #2: Yes

Reviewer #3: Yes

4. Is the manuscript presented in an intelligible fashion and written in standard English?

Reviewer #1: Yes

Reviewer #2: Yes

Reviewer #3: No

5. Review Comments to the Author

Reviewer #1: -The authors mentioned “Any disagreement that occurred during data extraction between researchers was handled 84 through discussion” What kind of disagreements? It needs explanation.

-One of the questions used in the checklist “Were the study participants sampled appropriately?” Most studies does not show how they calculated the sample size, they just mention the number of study participants. So how the authors addressed this?

-Did authors used a methods that confirms Klebsiella species is pathogenic i.e. not normal flora?

-Did the authors consider MDR definition used by different literatures? If so what definition they used?

-Discussion section: lower and higher prevalence discrepancies between studies should be explained (possible explanations for variations should be given).

-Conclusion section: Any suggestion or recommendations?

Reviewer #2: This systematic review and meta-analysis determines the prevalence of multidrug-resistant Klebsiella species using data collected between 2018 and 2022.

The study suggests a high prevalence of multidrug-resistant Klebsiella species in Ethiopia; prevalence varies regionally and according to the sample type analysed. The language is formal and objective, using appropriate subject-specific terminology. Furthermore, the text is grammatically correct and free from spelling and punctuation errors.

The study suggests a high prevalence of multidrug-resistant Klebsiella species in Ethiopia; prevalence varies regionally and according to the sample type analysed. Technical term abbreviations are explained when first used. Consistent citation and footnote styles are followed, adhering to conventional structure.

My comments on this review as follow:

1- The literature search and data collection were conducted appropriately, but the text and the presentation of figures do not correspond.

2- I understand that this is a review paper, and while the authors may wish to pay homage to the referenced articles, the figures appear cluttered, convoluted and difficult to understand with references to et al.

3- To compare the Ethiopian data with other countries, the authors could produce a single figure, which will be a good addition to the review.

Thank you so much to the authors for reviewing the prevelance of AMR resistance Klebsiella in Ethiopia.

Reviewer #3: 1. Title: Phenotypic and species must be added. Prevalence of Phenotypic Multi-drug resistant Klebsiella species recovered from clinical specimens in Ethiopia: A Systematic Review and Meta-analysis.

2. Introduction : revised line No 44 and 55 at the following positions:

44 communal flora: commensals flora

44 intestinal tract of humans and animals

55 for infections caused by these problematic species

3. Discussion: Not enough:

a. It is better to highlight the important of the study in the beginning of discussion.

b. More scientific explanation of the results is needed, such as possible causes of MDR occurrence.

c. References No 50, 51 and 52 talked about Klebsiella pneumoniae, so it is not true to mention Klebsiella spp there ? And author should mention Klebsiella pneumonia.

4. Study limitation: The author should to discuss the following limitations:

1. Absence of multi-drug resistance detection by molecular methods

2. The study period covered was short

6. PLOS authors have the option to publish the peer review history of their article (what does this mean?). If published, this will include your full peer review and any attached files.

Reviewer #1: **Yes: **Dr. Belayneh Regasa Dadi

Reviewer #2: No

Reviewer #3: **Yes: **Elhadi Abdalla Ahmed

---

## [Author Response · Author response to Decision Letter 0]

2 Dec 2023

We give a response for the comments with point-by-point response

---

## [Editor Report · Decision Letter 1]

2 Jan 2024

PONE-D-23-29785R1Prevalence of Phenotypic Multi-drug Resistant Klebsiella Species Recovered from Different Human Specimens in Ethiopia: A Systematic Review and Meta-analysisPLOS ONE

Dear Dr. Begashaw,

Thank you for submitting your manuscript to PLOS ONE. After careful consideration, we feel that it has merit but does not fully meet PLOS ONE’s publication criteria as it currently stands. Therefore, we invite you to submit a revised version of the manuscript that addresses the points raised during the review process.

We look forward to receiving your revised manuscript.

Kind regards,

Melaku Ashagrie Belete, MSc.

Academic Editor

PLOS ONE

Journal Requirements:

Additional Editor Comments:

The authors tried to modify the manuscript, however, there are still some unaddressed issues in the revision despite mentioned as modified in the point-by-point response. The newly added subgroup analysis is indeed vital and can now significantly show the probable differences in pooled estimates.

Please remove those columns with only 1 study included in the subgroup analysis because such data are once mentioned in your forest plot; for instance, Table 2: Somali regional state.

“However, this study revealed bias because of fifteen unpublished papers.” I don’t think this statement is correct as the inclusion of unpublished papers in SRMA cannot be a cause for bias. Please provide elaboration or make an amendment. Please make improvements to the statements under subsection ‘Publication bias’ as it sounds misleading.

Please remove Fig 5 and all the corresponding statements.

The subsection subgroup analysis is enough in your result. Please remove the subsections Subgroup analysis by Regional State, Subgroup analysis by specimen…. Besides, what was the significance of Table 2 if you mentioned all data in elaborative statements under the subsection? I suppose you should minimize the information in the elaborative statements as the Table is expressive by itself.

Add captions for your Supplementary files.

I can’t find all search strategies in detail in a supplement file in line with PRISMA recommendations (database, date searched, key words, number of articles retrieved etc.).

Table 4: Please change the heading of the 2nd column to ‘Study omitted’, remove the word omitting for the rest of the rows, and apply proper citation of the studies inside the table.

Please remove Supporting file 2 (zip file). The individual eligible studies are not required to be placed as a supporting file.

There are still plenty of typographical errors that need your immediate improvement.

Discussion 1st paragraph last phrase “antibiosis consumption” should be written as “antibiotic consumption”.

Figure 1: “Tile/abstract screened (n=81)” please correct it.

---

## [Author Response · Author response to Decision Letter 1]

3 Jan 2024

We included all the responses to the comments in point-by-point document as a word file.

---

## [Editor Report · Decision Letter 2]

5 Jan 2024

Prevalence of Phenotypic Multi-drug Resistant Klebsiella Species Recovered from Different Human Specimens in Ethiopia: A Systematic Review and Meta-analysis

PONE-D-23-29785R2

Dear Mr Tsegaye Alemayehu,

We’re pleased to inform you that your manuscript has been judged scientifically suitable for publication and will be formally accepted for publication once it meets all outstanding technical requirements.

Kind regards,

Melaku Ashagrie Belete, MSc.

Academic Editor

PLOS ONE
---

## [Editor Report · Acceptance letter]

2 Feb 2024

PONE-D-23-29785R2 

PLOS ONE

Dear Dr. Alemeyhu, 

I'm pleased to inform you that your manuscript has been deemed suitable for publication in PLOS ONE. Congratulations! Your manuscript is now being handed over to our production team.

Kind regards, 

on behalf of

Mr. Melaku Ashagrie Belete 

Academic Editor

PLOS ONE